# Unveiling the Role of Endothelial Dysfunction: A Possible Key to Enhancing Catheter Ablation Success in Atrial Fibrillation

**DOI:** 10.3390/ijms25042317

**Published:** 2024-02-15

**Authors:** George E. Zakynthinos, Vasiliki Tsolaki, Evangelos Oikonomou, Panteleimon Pantelidis, Ioannis Gialamas, Konstantinos Kalogeras, Epaminondas Zakynthinos, Manolis Vavuranakis, Gerasimos Siasos

**Affiliations:** 13rd Department of Cardiology, “Sotiria” Chest Diseases Hospital, Medical School, National and Kapodistrian University of Athens, 11527 Athens, Greece; boikono@gmail.com (E.O.); pan.g.pantelidis@gmail.com (P.P.); jyialamas@gmail.com (I.G.); kalogerask@yahoo.gr (K.K.); vavouranakis@gmail.com (M.V.); ger_sias@hotmail.com (G.S.); 2Critical Care Department, University Hospital of Larissa, Faculty of Medicine, University of Thessaly, Mezourlo, 41335 Larissa, Greece; vasotsolaki@yahoo.com (V.T.); ezakynth@yahoo.com (E.Z.); 3Cardiovascular Division, Brigham and Women’s Hospital, Harvard Medical School, Boston, MA 02115, USA

**Keywords:** endothelial dysfunction, atrial fibrillation, catheter ablation, endothelium, endothelial function

## Abstract

Atrial fibrillation, a prevalent type of arrhythmia, is increasingly contributing to the economic burden on healthcare systems. The development of innovative treatments, notably catheter ablation, has demonstrated both impressive and promising outcomes. However, these treatments have not yet fully replaced pharmaceutical approaches, primarily due to the relatively high incidence of atrial fibrillation recurrence post-procedure. Recent insights into endothelial dysfunction have shed light on its role in both the onset and progression of atrial fibrillation. This emerging understanding suggests that endothelial function might significantly influence the effectiveness of catheter ablation. Consequently, a deeper exploration into endothelial dynamics could potentially elevate the status of catheter ablation, positioning it as a primary treatment option for atrial fibrillation.

## 1. Introduction

Arrhythmias are a common phenomenon, often leading individuals to the emergency room. Among these, atrial fibrillation (AF), a common supraventricular arrhythmia, is currently estimated to have a prevalence ranging from 2 to 4%. With the increasing comorbidities associated with AF and the aging of the population, it is anticipated that the prevalence will surge by 2.3-fold in the coming years [1]. The lifetime risk of AF for Europeans is approaching nearly one in three individuals [1]. Consequently, the effective treatment of AF is of paramount importance.

Recent advancements in catheter ablation (CA) techniques for AF, including radiofrequency ablation (RF) and cryoablation, have shown promising results. However, there is still potential for improvement in these techniques.

The success of CA may be influenced by multiple factors. One such factor under investigation is the vascular endothelium. Covering a substantial surface area of the human body, the endothelium’s role in disease pathogenesis, especially in cardiovascular diseases, is well documented. Its importance in coronary disease is particularly significant. Moreover, recent studies have established a robust correlation between endothelial function and AF [2,3]. This raises the question of whether endothelial function could be correlated with the treatment of AF, and particularly with CA.

This review aims to clarify the connection between AF and endothelial dysfunction (ED) and to enhance our understanding of CA, including how endothelial factors might influence its success.

## 2. Endothelial Dysfunction and Atrial Fibrillation

### 2.1. Evaluating Indicators of Endothelial Dysfunction and Their Dynamics in the Context of Atrial Fibrillation

To understand the connection between the endothelium and atrial fibrillation (AF), it is essential to first define endothelial dysfunction and the methods used to assess endothelial function. The endothelium plays a crucial role in controlling blood fluidity, platelet aggregation, and vascular tone, and participates in immunological regulation, inflammation, and angiogenesis [4]. Endothelial function can be evaluated using circulating biomarkers and techniques measuring blood flow. The endothelium is vital for producing nitric oxide (NO), a key vasotransmitter. In endothelial cells, inactive NO synthase (NOS) becomes activated upon calcium entry through phosphorylation, converting L-Arginine into L-Citrulline and NO. NO’s primary function is to regulate vascular tone, and reduced NO levels indicate ED [5]. Asymmetric dimethylarginine (ADMA), a precursor and analog of L-Arginine, competitively inhibits NOS, so ADMA levels are also a marker for endothelial function assessment [6]. Elevated ADMA levels have been observed in patients with AF [7].

Von Willebrand factor (vWF), synthesized by endothelial cells, is a multimer produced and secreted upon the fusion of Weibel–Palade bodies with the cell membrane. It plays a critical role in stimulating platelet activation and aggregation, and its levels, commonly increased in AF, support the hypothesis of ED’s involvement in AF pathophysiology [8,9]. Additionally, endothelial cells express adhesion molecules like E-Selectins, P-Selectins, ICAM-1, and VCAM-1, which are involved in leukocyte recruitment and the inflammatory response [10]. These molecules, whose levels are generally higher in AF, can also be markers for ED evaluation, though studies utilizing adhesion molecules are limited [11,12].

Besides biomarkers, physiological measurement methods like flow-mediated dilation (FMD) are used for assessing endothelial dysfunction. FMD involves measuring brachial artery diameter using an ultrasound probe while a forearm cuff is inflated, inducing downstream tissue ischemia and vasodilation. After a 5 min inflation period, the cuff is deflated, and the resulting increase in brachial artery diameter indicates endothelial-dependent vasodilation. The sudden increase in blood flow upon cuff release causes enhanced endothelial shear stress and increased NO production in the brachial artery, leading to dilation [13]. Another method is reactive hyperemia–peripheral arterial tonometry (RH-PAT or RHI), which measures fingertip pulse amplitude increase following brachial pressure cuff inflation [14]. Generally, both methods show lower values in patients with AF [3,15].

### 2.2. The Impact of Electrical Cardioversion on Endothelial Function

The aforementioned evidence establishes a correlation between AF and ED. Supporting this hypothesis further is the observed improvement in endothelial function following cardioversion. A study involving thirty patients who underwent electrical cardioversion demonstrated a decrease in vWF levels four weeks post-cardioversion to sinus rhythm (SR), indicating a reduction in ED [16]. Additionally, an evaluation of ED post-electrical cardioversion using FMD revealed enhanced endothelial function, as evidenced by increased FMD values [17].

Supporting this, Skalidis et al. conducted a comparative study between healthy individuals with no history of AF and patients with AF who underwent electrical cardioversion. They assessed ED using FMD and nitroglycerin-mediated dilation (NMD). The findings showed that individuals with AF had lower FMD values than the control group. However, patients who reverted to and maintained SR exhibited significant improvements in FMD values both 24 h after cardioversion (8.0 ± 3.9% vs. 13.6 ± 5.3%, *p* < 0.001) and at the one-month mark (17.1 ± 3.9% vs. 17.2 ± 4.0% vs. 16.9 ± 4.1%). The FMD levels nearly matched those of healthy controls (95% confidence interval for mean difference −3.5 to 0.4, *p* = 0.11). Therefore, the close relationship between ED and AF not only underscores their interconnection but also suggests a potential role of ED in CA for AF [18].

## 3. Common Ground between Endothelial Function and Atrial Fibrillation

Atrial fibrillation and ED share numerous common risk factors, which likely explains their connection. Key risk factors for ED include hypertension, aging, smoking, hyperglycemia, and dyslipidemia (Figure 1) [19]. Similarly, most of these factors also contribute to the development of AF [20,21].

### 3.1. Hypertension’s Impact

Hypertension independently increases the risk of AF progression and accounts for about 14% of all AF cases [22]. It influences both the hemodynamic and electrical remodeling of the heart, as well as hormonal function and inflammation in the left atrium, thereby increasing the occurrence of AF [23]. Moreover, the activation of the renin–angiotensin–aldosterone system (RAAS) escalates during hypertension, which, as animal studies have shown, promotes atrial fibrosis due to the effects of angiotensin II [24,25]. There appears to be a bidirectional relationship between ED and hypertension: hypertension induces an inflammatory state with elevated reactive oxygen species (ROS), causing ED, while high microvascular pressure accelerates endothelial cell aging and turnover [26,27]. Conversely, ED can lead to hypertension due to an imbalance of endothelium-derived relaxing and contracting factors, including a limited production of NO, resulting in vasoconstriction [27]. This mutual relationship is further supported by the improvement of endothelial function with successful pressure control using angiotensin receptor blockers and angiotensin-converting enzyme inhibitors [28,29]. Therefore, ED may influence the occurrence of AF through its impact on hypertension development.

### 3.2. Ageing and Its Role

Additionally, mechanisms such as senescent cells in the vascular system, mitochondrial dysfunction, ROS overproduction, chronic inflammation, the loss of proteostasis, the exhaustion and/or dysfunction of stem/progenitor cells, the dysregulation of certain microRNAs (miRNAs), and extracellular matrix remodeling contribute to progressive vascular aging and indirectly to endothelial dysfunction [30]. Similarly, advanced age leads to extensive atrial fibrosis, a key mechanism in the development of AF [31].

This is underscored by the fact that atrial fibrillation stands out as the most prevalent arrhythmia, with its occurrence varying from 0.1% in patients under 55 years to more than 9% in octogenarians [32]. Consequently, aging emerges as a significant factor in the occurrence of AF. However, this population also exhibits a high incidence of other comorbidities, such as diabetes and heart failure, both proven to impact endothelial function [33]. To adequately evaluate and comprehensively treat these patients, it is imperative to further elucidate the intricate association of different comorbidities with endothelial function.

### 3.3. Diabetes and Its Influence

In diabetic patients, diabetic cardiomyopathy is characterized by diastolic dysfunction, similar to the effects of hypertension, and is exacerbated by an increased production of advanced glycation end products (AGEs) and their receptors, leading to fibrous scarring of the left atrium [34,35]. This creates conducive conditions for AF [36]. In the context of AF development in the heart, increased sympathetic activity, triggered by parasympathetic denervation, results in an imbalance in the autonomic nervous system. This is particularly noteworthy in diabetic patients, where heightened sympathetic tone stems from the initial denervation of the parasympathetic system, creating a state of sympathetic hyperactivity [37]. Diabetes has been linked to the abnormal expression of connexins (gap junction proteins) [38], leading to delayed conduction velocities, intra-atrial electromechanical delay, prolonged action potentials, and abnormal atrial voltage [36,39,40]. Diabetes also impairs endothelial function in both small and large vessels through multiple pathophysiological pathways [41], including reduced NO production, oxidative stress, impaired endothelial repair, and an imbalance of vascular endothelial growth factor (VEGF) and NO. These effects are due to the reduced protein expression of endothelial nitric oxide synthase (eNOS), caused by the reduced phosphorylation of its active site and the enhanced phosphorylation of its inhibitory site [42,43]. Mitochondrial dysfunction, the increased activity of NADPH oxidase, cyclooxygenase, and other enzymes, and decreased superoxide dismutase contribute to oxidative stress [44,45,46]. Pathological endothelial repair is caused by the reduced circulation of progenitor and bone-marrow-derived hematopoietic cells in individuals with diabetes [47,48], and endothelial progenitor cells in individuals with diabetes lose their autocrine and paracrine functions, further reducing cell repair [49]. Furthermore, VEGF levels are elevated in individuals with diabetes, but show variable expression in different tissues [50]. VEGF and NO, which normally cooperate in angiogenesis, become imbalanced, negatively affecting angiogenesis and the production of functional endothelium [51].

### 3.4. Lipid Profile’s Significance

Dyslipidemia has not been definitively linked to an increased incidence of AF. Data from the Multi-Ethnic Study of Atherosclerosis and the Framingham Heart Study suggest a protective role of high-density lipoproteins (HDLs) against AF and an increased risk associated with elevated triglycerides. Conversely, low-density lipoproteins (LDLs) and total cholesterol do not appear to significantly affect AF prevalence [52]. Notably, patients with AF have plasma triglyceride levels that are 2.6 times higher than those without AF [53]. Elevated blood lipid levels can create an inflammatory environment and increase oxidative stress [54], which may contribute to the development of AF. The impact of lipids on endothelial function is well recognized. Although the detrimental effects of LDLs are known, recent research from Japan indicates optimal FMD values within a specific LDL-C range of 71–100 mg/dL, with FMD values declining outside this range [55]. Higashi’s review on the role of lipids in endothelial function revealed a reverse U-shaped relationship between HDL-C levels and FMD, and a linear inverse correlation between triglyceride levels and FMD in individuals not undergoing lipid-lowering therapy [56]. Oxidized LDLs suppress NO production, promote leukocyte adhesion, enhance smooth muscle cell growth factor activity, inhibit thrombomodulin production, and trigger apoptosis [56]. Endothelial cell membranes have scavenger receptors, such as lectin-like oxidized LDL receptor-1 (LOX-1) and scavenger receptor class A (SR-A), which absorb oxidized LDLs and contribute to apoptosis [57,58]. Furthermore, high LDL-C levels increase Rho-associated kinase activity, destabilizing eNOS mRNA and reducing NO production, thereby contributing to ED [59,60]. Conversely, HDLs have a positive effect on endothelial function, chiefly by inhibiting LDL oxidation, reducing stress, preventing apoptosis, and activating eNOS [61,62,63]. The intricate connection between lipids and ED, along with the underlying mechanisms, is a complex subject that has been extensively studied. However, more surveys are needed to further elucidate the correlation between AF and ED.

### 3.5. Smoking’s Contribution

Finally, smoking, similar to ED, was identified as a risk factor for AF in the Atherosclerosis Risk in Communities (ARIC) study [64]. Further analysis from the ARIC study revealed a dose-dependent relationship between tobacco use and AF incidence, with active smokers showing a stronger correlation with AF risk compared to former smokers. Smoking induces significant vascular inflammation, characterized by a surge in cytokines and inflammatory cells, compromising endothelial integrity and exacerbating oxidative stress. Intriguingly, even newer devices like electronic cigarettes may pose risks to the endothelium and should be avoided [65].

## 4. Catheter Ablation for Atrial Fibrillation and Endothelial Dysfunction

Catheter ablation for the treatment of AF has significantly advanced in recent years, and its clinical benefit in reducing AF-related symptoms is well established. Moreover, for patients with heart failure and reduced ejection fraction, catheter ablation has been shown to reduce all-cause mortality and hospitalization rates. However, CA is still considered a second-line treatment for AF, with antiarrhythmic drugs (AAD) remaining the primary choice for maintaining SR [1]. The preference for AAD over CA is partly due to the majority of clinical trials not yet demonstrating CA’s superiority over AAD in terms of quality of life (QoL) [66] and complication rates [67,68], as well as the considerable risk of AF recurrence post-ablation [1]. In contrast to previous data, a recent study comparing cryoablation with AADs demonstrated the superiority of cryoballoon ablation, underscoring the pivotal role of cryoablation in the treatment of AF and highlighting the necessity to further improve its success rate [69]. Therefore, identifying and modifying potential risk factors for CA failure is crucial to enhance its success rate. The connection between endothelial function and AF is discussed earlier in this review. However, the potential impact of endothelial function on the success of CA remains unclear. It is possible that a reciprocal relationship exists between CA and endothelial function.

### 4.1. The Role of Endothelial Dysfunction in Catheter Ablation

Endothelial function appears to be a crucial factor in the success of CA for AF, as suggested by several studies (Table 1). Research evaluating endothelial function through biochemical markers like Endothelin-1 (ET-1) or mechanical parameters such as FMD has established a correlation between the state of the endothelium before CA and its success. Studies have consistently shown that ET-1 levels are higher in patients with persistent AF (PeAF) or paroxysmal AF (PAF) compared to healthy individuals [70,71,72]. Additionally, lower ET-1 levels were associated with a greater likelihood of maintaining AF-free status three months post-ablation [71], and ET-1 was identified as an independent risk factor for the postoperative recurrence of PeAF only [70]. However, Wang et al. recognized ET-1 as a predictor of AF recurrence, particularly in PAF patients, contrasting with other studies where the main predictor for PeAF recurrence was the duration of AF [72]. They also found a positive correlation between ET-1 levels and increased left atrium diameter (r = 0.18, *p* = 0.02), and noted that patients with lower ET-1 levels (below a certain cut-off) were less likely to experience new AF episodes post-CA (40.85% vs. 59.15%, *p* = 0.012).

Similarly, ADMA levels are higher in AF patients compared to control groups, and elevated serum ADMA levels have been linked to increased AF recurrence. The adjusted hazard ratio (HR) for patients in the upper 50th percentile of ADMA levels, compared to the lower 50th percentile, was 4.59 (95% CI, 1.81–11.62; *p* = 0.001). Reactive hyperemia–peripheral arterial tonometry (RH-PAT or RHI) as an alternative method for assessing endothelial function also yielded comparable results. The baseline RHI in AF patients was significantly lower than in non-AF individuals (0.70 ± 0.24 versus 0.50 ± 0.23, *p* < 0.001) [73]. Patients with endothelial dysfunction, as indicated by RHI, tended to have a higher incidence of cardiovascular events, not limited to AF episodes [74]. Patients with either PAF or PeAF had lower LnRHI values than healthy controls [73]. Moreover, 3–6 months post-CA, patients with AF recurrence demonstrated lower LnRHI values (3 months: 0.69 ± 0.23 vs. 0.78 ± 0.25, *p* = 0.045; 6 months: 0.66 ± 0.19 vs. 0.74 ± 0.22, *p* = 0.044), and the improvement in LnRHI 3 months post-CA was less pronounced in the recurrence group compared to the non-recurrence group (−0.09 ± 0.26 vs. 0.17 ± 0.27, *p* < 0.001) [73]. Recently, Okawa et al. found that baseline ED (assessed by RHI) did not increase AF recurrence rates [74], yet a Japanese study identified LnRHI values 3 months post-CA as an independent marker of AF recurrence, with high sensitivity and specificity. They also observed that a decrease in LnRHI 3 months post-CA, compared to baseline, significantly predicted AF recurrence [73].

Consequently, endothelial dysfunction may play a significant role in the success of CA, and it might be beneficial to assess endothelial function in patients prior to undergoing CA.

**Table 1 ijms-25-02317-t001:** Studies showing the effect of endothelial dysfunction on the recurrence of atrial fibrillation after catheter ablation.

Studies	Design	Total Cases—Follow Up	Population	Intervention—ED Assessment	Findings
Okawa et al. [74], 2023	Prospective Cohort Study	1040 with AF—Median: 35 months	Japanese (2013–2022) (mean age 67 ± 10)	RF-RHI	Higher 5-year incidence of cardiovascular events in the ED group vs. non-ED group: 98 (11.8%) vs. 13 (6.2%). AF recurrence in ED HR 1.01, 95% CI (0.68–1.5), *p* = 0.86.
Gao et al. [70], 2022	Case Control	66 PeAF, 72 PAF, 80 control—6 months	Chinese	RF-ET-1 levels, CTGF	Higher levels of ET-1 and CTGF in PAF and PeAF, compared to control. Higher levels of ET-1 and CTGF in patients with postoperative AF recurrence than those without. Positive correlation of ET-1 and CTGF levels pre- and postoperatively, with PeAF recurrence.
Kanazawa et al. [73], 2021	Retrospective Observational Study	214 (151 SR, 63 AF)—12 months	Japanese (2013–2016) (mean 61 ± 10)	RF-RH-PAT	LnRHI 3 months after CA (decreased to ≥0.01 compared with that before CA) was an independent marker of suspected AF recurrence (sensitivity, 0.806; specificity, 0.821; area under the curve, 0.792; *p* < 0.001). Higher probability of AF recurrence when the LnRHI value 3 months after CA was lower than that before CA (log rank test, *p* < 0.001).
Lackermair et al. [71], 2017	Case Control	96 AF, 40 Control—3 months	German (61.8 ± 10.9 AF, 60.2 ± 12.58 Control)	RF-ET-1, CGA, MCP-1	Higher levels of ET-1 in patients with AF, compared to age- and sex-matched healthy volunteers without AF (2.62 pg/mL vs. 1.57 pg/mL; *p* < 0.001). Lower ET-1 levels prior ablation associated with freedom of AF in the follow-up period of 3 months (2.57 pg/mL vs. 2.96 pg/mL; *p* = 0.02) (MCP-1 plasma levels increased significantly after ablation independent from AF recurrence; CGA levels increased significantly only in patients without recurrence towards the level of healthy controls).
Matsuzawa et al. [75], 2016	Double-Blind, Placebo-Controlled Trial	92 (enrolled, 71 follow up)—3 months	American, (January 2008 and December 2009), (mean 57 ± 10)	RF-RH-PAT	Association of Ln_RHI levels with symptomatic AF (hazard ratio [HR] 1.99 [95% CI 0.92–4.51], *p* = 0.079) and atrial arrhythmia recurrence (HR 1.93 [95% CI 0.99–3.92], *p* = 0.054); ≤60 years + attenuated endothelial function significantly associated with increased risk of symptomatic AF recurrence (HR 4.01 [95% CI 1.39–14.38], *p* = 0.009). No significant association in participants aged >60 years. Endothelial dysfunction → higher rates of recurrence of AF (*p* = 0.010).
Wang et al. [72], 2012	Prospective Cohort Study	103 PAF, 55 PeAF—22-month median follow up	Chinese	RF-big ET-1	Higher plasma levels of big ET-1 in the recurrence group vs. in the non-recurrence group in all patients (*p* = 0.001). ET-1 levels were a prognostic predictor of AF recurrence only in patients with paroxysmal AF (*p* = 0.037).
Yang et al. [76], 2011	Prospective Cohort Study	138 with AF—3 months	Chinese (June 2007 to October 2009), (49.84 ± 7.47 AF rec, 50.02 ± 6.93 no rec)	RF-ADMA	Higher serum ADMA concentrations before catheter ablation in the recurrence group (0.75 ± 0.24 vs. 0.58 ± 0.14 μmol/L; *p* < 0.001). Association of recurrences of AF with higher serum ADMA concentration (HR = 4.42; 95% CI, 1.93–10.12; *p* < 0.001).

Asymmetric dimethylarginine (ADMA), connective tissue growth factor (CTGF), Chromogranin A (CGA), Endothelin-1 (ET-1), Monocyte chemotactic protein-1 (MCP-1), paroxysmal atrial fibrillation (PAF), persistent atrial fibrillation (PeAF), Peripheral vascular reactive hyperemia index (RHI), radiofrequency (RF), reactive hyperemia–peripheral arterial tonometry (RH-PAT).

### 4.2. The Role of Catheter Ablation on Endothelial Dysfunction

#### 4.2.1. Periprocedural and Immediately Postprocedural Role

Catheter ablation is an invasive procedure that utilizes various energy forms, such as radiofrequency, cryotherapy (extreme cold), and electrical pulses, to isolate the pulmonary veins. This energy transfer disrupts the normal function of cardiac tissues, consequently affecting the endothelium of heart vessels both periprocedurally and immediately afterward. Studies on radiofrequency ablation in porcine hearts have shown that applying RF energy near coronary arteries (less than 1 mm) results in defective endothelium-dependent contractions and relaxations, indicative of endothelial dysfunction (ED). Histopathological examinations of porcine coronary arteries revealed vascular wall disruption when RF was applied within 5 mm of the artery [77]. However, most studies assessing CA’s impact on endothelial function rely on indirect markers, as direct histopathological data are scarce. Commonly used markers include the von Willebrand factor (vWF), which is observed to elevate immediately after RF energy application and remains elevated for at least 24 h post-procedure [78]. Concurrently, RF may also influence the coagulation system, as evidenced by increased tissue plasminogen activator (t-PA) and D-Dimers (DD), and decreased plasminogen activator inhibitor-1 (PAI-1) levels [79], potentially creating a thrombogenic state. Elevated vWF levels persist for at least 48 h, and some studies suggest persistence even up to 72 h post-procedure [79,80]. This sustained elevation occurs despite the reduction in other biomarkers like sP-Selectin that also rise periprocedurally [79]. Similarly, plasma NO levels significantly decrease during the procedure [81,82], reinforcing the hypothesis of ED during RF ablation. Lim et al. correlated NO levels with the microvascular resistance index and other molecules, such as activated leukocyte cell adhesion molecule (ALCAM) and lipoprotein-associated phospholipase (LpPLA2), and observed general coronary microvascular dysfunction immediately after ablation [81]. The duration of this dysfunction, however, remains undefined.

These findings pertain to RF ablation and do not necessarily extend to newer techniques like cryoablation or pulse field ablation. Cryoablation exhibits a similar endothelial damage profile (based on vWF levels) and coagulation activation (based on sP-Selectins) to RF [83,84]. However, cryoablation appears to cause more extensive myocardial damage, indicated by significantly higher troponin levels [83]. When evaluating ED using the L-arginine/ADMA ratio, cryoablation was less detrimental than RF. Both methods had similar baseline ratios, but the cryoablation group showed a significant increase post-procedure (*p* = 0.02), indicating a safer profile for cryoablation regarding ED [85]. Hajas et al. compared cryoablation, a pulmonary vein ablation catheter (PVAC-Gold), and irrigated radiofrequency (IRF) for endothelial cell damage using three different ED biomarkers (vWF, Factor VIII (FVIII), and VCAM). All three contemporary ablation techniques caused significant endothelial damage, with increased levels of vWF and FVIII across all methods and elevated VCAM in cryoablation and IRF (but not in PVAC) [86]. Pulse field ablation, on the other hand, did not significantly differ from RF in terms of vWF levels, but showed milder effects on platelet activation and the coagulation cascade. However, pulse field ablation has been shown to induce lower levels of inflammation compared to conventional energy sources. This, combined with the reduced platelet activation associated with pulse field ablation, suggests that this ablation method may be more conducive to endothelial function in the short term. Nevertheless, it revealed substantially higher myocardial damage than RF [87]. Overall, while short-term pulse field ablation may be more beneficial for ED, the long-term effects are likely to be similar to those of other CA methods. However, as of now, only a few studies exist, and more prospective research is needed to validate this hypothesis.

Overall, all ablation techniques induce temporary ED, the precise duration of which remains to be determined. Identifying the exact post-procedural period of ED is crucial for assessing complication risks and mitigating them. Moreover, the reliance on biomarkers with uncertain reliability and clinical significance necessitates further research incorporating histopathological and imaging data to fully understand the relationship between ED and CA.

#### 4.2.2. Long-Term Role

In addition to its immediate impact, catheter ablation (CA) plays a significant long-term role in endothelial function, as demonstrated by various studies (Table 2). Endothelin-1 levels were initially lower in healthy controls compared to AF patients. Three months post-CA, patients without new AF episodes exhibited a drop in ET-1 levels, nearly matching those of healthy individuals. In contrast, patients with AF recurrence showed unchanged ET-1 levels three months after CA [71]. However, other studies did not confirm these findings, showing no significant difference in ET-1 levels between the no-recurrence and recurrence groups (25.5 ± 4.4 vs. 26.7 ± 5.4; *p* = 0.286) [88]. The results regarding ADMA levels were also mixed. One study of 101 AF patients undergoing CA reported unchanged ADMA levels before and six months after CA, regardless of AF recurrence [89]. Conversely, Lim et al. observed a decrease in endothelial dysfunction, as measured by ADMA levels six months post-ablation in the SR maintenance group, but not in the AF recurrence group [90]. This group also measured platelet activation levels using P-selectin (CD62P) and PAC-1 markers. Similar to ADMA, unchanged levels of both biomarkers were seen in the AF recurrence group six months post-CA, whereas the SR maintenance group showed decreased platelet activation (CD62P and PAC-1 levels) [90]. These results indicate that CA affects not only endothelial function but also the coagulation cascade. A Japanese study assessed endothelial dysfunction using levels of plasminogen activator inhibitor-1 (PAI-1) and soluble thrombomodulin (s-TM), finding increased levels of both markers six months post-CA, regardless of AF recurrence [89]. This suggests a positive impact of CA on endothelial function, possibly due to longer periods of AF freedom in all patients. However, more research is needed to solidify this hypothesis.

Nitric oxide levels also increased post-CA in AF patients, though they did not reach the levels of AF-free controls. Immunohistochemistry assays evaluating NO production and eNOS expression showed downregulation in AF patients, which CA was able to reverse [91]. Recent studies have shifted focus to microRNAs (miRNAs) due to their relationship with NOS. Research on swine models identified an increased expression of miR-155-5p and miR-24-3p in AF patients, which decreased post-CA. This correlation, along with NOS levels, confirmed the involvement of these miRNAs in AF pathophysiology [91]. Additionally, the direct inhibition of miR-24-3p and miR-155-5p on eNOS was noted, suggesting their role in AF recovery post-CA [90]. Namino et al. investigated different miRNAs in humans, including miR-22, miR-126, and miR-142, finding correlations between miR-22, miR-126 levels, and levels of ADMA and s-TM, respectively. They also noted unchanged values of miR-22 and miR-142, and a decrease in miR-126 six months post-CA [89].

Endothelial dysfunction was further assessed using reactive RHI and FMD in multiple studies, showing overall improvement with CA. Both RHI and FMD methods indicated increased endothelial function six months post-CA in patients who maintained SR, but not in those with AF recurrence [73,88]. This improvement was observed from the first day after ablation in patients without AF recurrence [92]. Interestingly, a Japanese study with 102 patients with paroxysmal AF (PAF) and 75 with persistent AF (PeAF) reported an improvement in LnRHI after six and twelve months only in patients who recurred from PeAF, not those with baseline PAF [93]. However, an American study with 92 AF participants did not show significant ED improvement post-CA, except in patients treated with atorvastatin for dyslipidemia, albeit without statistical significance [75].

In summary, apart from a brief period of dysfunction observed in the days following CA, endothelial function generally shows improvement in AF patients, especially in those who sustain SR for an extended period thereafter. Patients with new AF episodes do not exhibit similar improvements, underscoring the importance of SR maintenance post-CA, potentially through a combination of CA and drug therapy.

**Table 2 ijms-25-02317-t002:** Studies showing the effect of catheter ablation on endothelial dysfunction.

Studies	Design	Total Cases—Follow Up	Population (Period), (Age)	Intervention—ED Assessment	Findings
Kanazawa et al. [73], 2021	Retrospective Observational Study	214 (151 SR, 63 AF)—12 months	Japanese (2013–2016) (mean 61 ± 10)	RF-RH-PAT	Improvement in LnRHI 6 months after CA (0.61 ± 0.25 versus 0.74 ± 0.22, *p* < 0.001) in patients without AF recurrence, but not in patients with AF recurrence (0.78 ± 0.25 versus 0.66 ± 0.19, *p* = 0.055). LnRHI in AF rhythm before CA remained unchanged 6 months after CA. Improvement in LnRHI in AF SR group 3 and 6 months after CA compared with that before (3 months: 0.66 ± 0.24 versus 0.81 ± 0.25, *p* < 0.001; 6 months: 0.66 ± 0.24 versus 0.76 ± 0.23, *p* = 0.012). Improvement in LnRHI in patients without AF recurrence after CA to a level similar to those in normal control.
Namino et al. [89], 2019	Prospective Cohort Study	101–6 months	Japanese (November 2014–August 2015) (mean 61.8 + 8.6)	RG-PAI-1, s-TM, ADMA	Increase after catheter ablation of s-TM and PAI-1 levels at the 6-month follow up compared with baseline in both the restored SR and recurrent AF groups (11.55 [2.92] vs. 13.75 [3.38], *p* < 0.001; 10.28 [2.78] vs. 11.67 [3.37], *p* < 0.001) and (25.74 [15.25] vs. 37.79 [19.56], *p* < 0.001; 26.16 [15.70] vs. 40.74 [22.55], *p* < 0.001), respectively. No differences in ADMA levels at the 6-month follow up compared with the baseline for either group ((0.625 [0.163] vs. 0.589 [0.101], *p* = 0.241) and (0.637 [0.143] vs. 0.616 [0.102], *p* = 0.500)).
Wang et al. [91], 2019	Case Control	20 PAF, 20 control (+pigs)	Chinese (52.5 ± 9.3 AF, 53 ± 10.8)	RF-miRNA	miR-99b-3p, miR-133a, and miR-99b expression reduced by almost 75%—expression of miR-325, miR-423-5p, and miR-451a reduced by 25% post-ablation. Decreased levels of NO in AF+ groups (both pre-ablation and post-ablation). Implication of miR-155, miR-24, and eNOS on AF pathogenesis (on pigs).
Lackermair et al. [71], 2017	Case Control	96 AF, 40 Control—3 months	German (61.8 ± 10.9 AF, 60.2 ± 12.58 control)	RF-ET-1 CGA, MCP-1	Patients without AF recurrence demonstrated a decrease in ET-1 levels three months after ablation getting closer to the level of the healthy volunteers (2.33 pg/mL vs. 2.57 pg/mL; *p* < 0.01), whereas ET-1 levels in patients with AF recurrence remained unchanged at an elevated level (2.83 pg/mL vs. 2.96 pg/mL; *p* = 0.09). MCP-1 plasma levels increased after ablation independent from AF recurrence; CGA levels increased significantly only in patients without recurrence towards the level of healthy controls, but not in patients with recurrence.
Okawa et al. [93], 2017	Case Control	102 PAF, 75 PeAF, 51 control	Japanese (May 2013 to February 2015) (65.9 ± 10.0 PAF, 65.8 ± 10.7 PeAF, 64.9 ± 10.8 control)	RF-RH-PAT	Lowest RHI in the PeAF group (*p* < 0.001 versus control, *p* = 0.008 versus PAF groups). Unchanged RHI measurements in the PAF patients prior to the catheter ablation and at 6 and 12 months post-ablation. Increased RHI at the 6-month follow up in the PeAF group (0.53 ± 0.28, *p* < 0.05), which was maintained at 12 months.
Matsuzawa et al. [75], 2016	Double-Blind, Placebo-Controlled Trial	92 (enrolled, 71 follow up)—3 months	American, (January 2008 and December 2009), (mean 57 ± 10)	RF-RH-PAT	Unchanged endothelial function after atrial ablation (Ln_RHI from 0.60 ± 0.29 to 0.65 ± 0.25, *p* = 0.41) after 3 months. Slightly higher endothelial function in the atorvastatin group than in the placebo group (not statistically significant).
Lim et al. [90], 2014	Prospective Cohort Study	57 AF patients—6 months	Australian (53.8 ± 10.5, SR maintenance, 61.2 ± 9.4, AF recurrence)	RF-ADMA and platelet activation receptors CD62P (P-selectin) and glycoprotein IIb/IIIa (PAC-1)	After catheter ablation and successful maintenance of SR, endothelial dysfunction measured by ADMA levels decreased at 6-month follow up compared with baseline (log ADMA μM/L 0.15 ± 0.02 vs. 0.17 ± 0.04, *p* = 0.015). No significant improvement in ADMA levels was seen in the group that sustained AF recurrence (log ADMA μM/L 0.16 ± 0.03 vs. 0.16 ± 0.02, *p* = 0.4).
Yoshino et al. [92], 2013	Case Control	48 (27 AF, 21 control)—6 month	Japanese (April 2008 to August 2010) (58 ± 12 AF, 56 ± 17 control)	RF-RH-PAT	Higher log_e_ RHI the morning after ABL, compared with that before ABL in day 1-restored SR group (0.53 ± 0.20; 0.73 ± 0.25; *p* = 0.009), which was maintained after 6 months, and no difference in the day 1-recurred AF group (0.49 ± 0.21; 0.52 ± 0.23; *p* = 0.787). Similar log_e_ in day 1-recurred AF group for all points (before ABL, day after, and 6 months after).
Shin et al. [88], 2011	Prospective Cohort Study	61 PAF, 19 PeAF, 80 control—6 months	Korean (53.4 ± 10.4 AF, 54.3 ± 9.3 Control)	RF-FMD + ET-1	Lower FMD baseline in younger AF patients than control. Greater FMD in AF subjects who remained in SR after a successful CA at 1-month post-CA when compared with FMD baseline, and even more significant increases 6 months post-CA, nearly approaching control levels. Lower FMD baseline in AF recurrence group compared with nonrecurrence group, without increase for 1 month post-CA, even though SR was maintained.

Asymmetric dimethylarginine (ADMA), connective tissue growth factor (CTGF), Chromogranin A (CGA), Endothelin-1 (ET-1), Monocyte chemotactic protein-1 MCP-1, paroxysmal atrial fibrillation (PAF), persistent atrial fibrillation (PeAF), Peripheral vascular reactive hyperemia index (RHI), plasminogen activator inhibitor-1(PAI-1), soluble thrombomodulin (s-TM), radiofrequency (RF), reactive hyperemia–peripheral arterial tonometry (RH-PAT), flow-mediated dilatation (FMD).

## 5. Catheter Ablation and Endothelial Function Treatment

### 5.1. Treatment of Risk Factors and Lifestyle Interventions

A baseline of good endothelial function enhances the outcomes of catheter ablation, increasing the likelihood of sustained sinus rhythm. Moreover, maintaining sinus rhythm post-ablation contributes to a gradual improvement in ED. Consequently, prolonged sinus rhythm not only enhances endothelial function but also diminishes the likelihood of atrial fibrillation recurrence, creating a cyclical pattern of positive outcomes. Thus, a holistic approach is crucial when treating a patient with AF and ED who is undergoing CA. Lifestyle interventions like weight loss and smoking cessation, along with an improved control of cardiovascular risk factors, have been shown to enhance endothelial function and reduce AF recurrence.

Considering the role of hypertension, this emerges as a common risk factor in the intricate interplay between AF and ED. A meta-analysis involving 56,308 patients with diastolic dysfunction or left ventricular hypertrophy, treated with RAAS inhibitors, demonstrated up to a 40% reduction in AF incidence due to better arterial pressure control [94]. Similarly, treating hypertensive patients with RAAS inhibitors led to improvements in FMD values and, consequently, endothelial function [28,29].

Shifting our focus to glycemic control, its pivotal role in AF management comes to the forefront. A 10% reduction in HbA1c levels 12 months before CA significantly decreases the likelihood of AF recurrence [95]. Ideally, maintaining HbA1c levels below 6.9% increases the chances of successful ablation [96]. Specific drugs like biguanides, thiazolidinediones, secretagogues, and SGLT-2 inhibitors have shown positive effects in reducing the risk of AF recurrence [97]. Improved glycemic control also enhances endothelial function. Patients with type II diabetes mellitus (DMII) treated with SGLT-2 inhibitors showed substantial improvement in FMD values and endothelial function [98]. Additionally, increased daily exercise improved ED in patients with DMII [99].

While the precise role of lipids in AF pathophysiology remains unclear [19], possibly owing to insufficient surveys, effective lipid control is pivotal in managing AF episodes, with statins playing a key role [100,101]. The Justification for the Use of Statins in Prevention: An Intervention Trial Evaluating Rosuvastatin (JUPITER) trial showed that rosuvastatin, compared to placebo, reduced the risk of AF [102]. Overall, statin use significantly lowers the incidence of AF recurrence post-CA (OR 0.81, 95% CI 0.59–1.10, *p* = 0.18) [103]. The role of lipid control in endothelial function, particularly in coronary and peripheral artery disease, is well established. A large meta-analysis demonstrated significant improvement in endothelial function with statins (standardized mean difference (SMD) 0.66, 95% CI 0.46–0.85, *p* < 0.001), not only in coronary circulation but also in peripheral circulation [104].

Lifestyle interventions are also crucial. Williams et al. showed that a mean weight reduction of 23.4 kg resulted in an improvement in FMD from 5.3% to 10.2% [105]. The LEGACY study, which divided AF patients into three groups based on weight loss, found that the frequency, severity, and duration of AF significantly improved in the two groups with more weight loss (*p* < 0.001), with a lower rate of AF recurrence in these groups [106].

In the context of the ARREST-AF Cohort Study, the study highlights the importance of aggressively managing multiple risk factors, leading to enhanced ablation success rates and increased arrhythmia-free survival [107]. Effective risk factor management not only decreases the need for initial ablation but also reduces the necessity for redo ablation procedures [108], emphasizing the benefits of a comprehensive approach for patients undergoing CA.

### 5.2. Antiarrhythmic Drugs after Catheter Ablation

Currently, the protocol for using antiarrhythmic drugs (AAD) post-CA remains somewhat ambiguous. Guidelines lack specific directives regarding the continuation or tapering of AADs, generally recommending their use for a duration ranging from 6 weeks to 3 months to minimize AF recurrence [1]. Large trials have indicated a reduction in AF recurrence, hospitalizations, and the need for cardioversions with the use of AADs for about 3 months following CA [109,110]. Interestingly, the use of AADs that were previously ineffective has also been shown to reduce post-CA AF recurrence [111]. Guidelines mention a ‘blanking period,’ after which the continued use of AADs can reduce AF recurrence. Trials cited earlier in this review have highlighted the improvement in endothelial function and its correlation with AF recurrence, suggesting that the blanking period might be related to endothelial function. However, this aspect has not been included in trials examining AADs after ablation. Therefore, ED may be a key factor in understanding and effectively managing AF recurrence. It may be prudent to assess ED post-CA to inform decisions regarding the weaning or continuation of AADs based on ED status. A suggested approach for this is illustrated in Figure 2. Nonetheless, a substantial number of trials are needed to fully evaluate and validate this hypothesis.

The optimal method for the perioperative evaluation of endothelial function remains an unresolved question. A comprehensive comparison of various methods for endothelial dysfunction (ED) assessment revealed that no single test can serve as a surrogate for another [112]. Notably, a limitation of this review was the exclusion of biomarkers, which were neither included nor compared for their effectiveness in evaluating ED. Consequently, it is recommended to adopt a blended approach, utilizing both practical methods and biomarkers for ED assessment. The choice between these approaches can be based on factors such as the invasiveness of the procedure and the resources available at each medical center.

## 6. Key Points

Patients undergoing catheter ablation should undergo a preoperative evaluation of endothelial function, as well as follow-up assessments for a period after the procedure. The evaluation of endothelial function, both pre- and postoperative, should encompass a dual approach involving biomarkers and practical methods like FMD and RH-PAT for a thorough assessment. However, it is imperative to note that further studies are required to precisely determine the most effective methods and the optimal time periods for their application.Endothelial function should be monitored in patients with atrial fibrillation, as it may significantly influence the treatment decisions for these patients.A comprehensive approach is essential for patients with atrial fibrillation and endothelial dysfunction. This includes managing common risk factors shared by both conditions to effectively treat the patients and minimize potential complications.

## 7. Conclusions

Endothelial function is a complex and critical factor in cardiovascular disease. Similarly, atrial fibrillation is a poorly understood condition, making its treatment challenging. Recent data highlight a significant overlap between these two entities, suggesting a need for an integrated approach to their treatment. Additionally, there is growing evidence supporting the influence of endothelial function on the success of catheter ablation for atrial fibrillation. Consequently, there is a pressing need for randomized clinical trials to elucidate the role of endothelial function and improve the success rates of catheter ablation. Potentially, innovative techniques like cryoablation may emerge as key factors in further enhancing ED outcomes in the future. Further studies are essential to understand their impact on ED in the months following CA. Furthermore, additional research is imperative to elucidate the precise postoperative time frame for evaluation and identify the most effective options for assessment. It is conceivable that a synergistic combination of various methods may be required for a comprehensive and accurate evaluation of endothelial function. It is noteworthy that the evolution of new techniques in the future could potentially reshape the landscape, bringing about revolutionary changes.

## Figures and Tables

**Figure 1 ijms-25-02317-f001:**
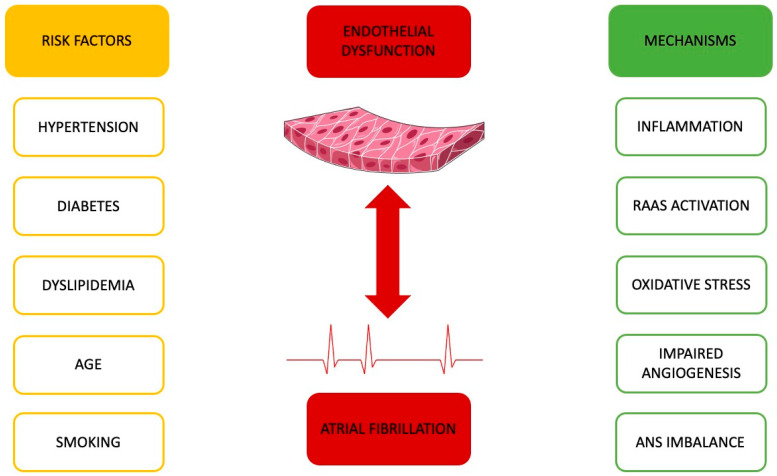
Shared risk factors for atrial fibrillation and endothelial dysfunction (**left**). Key mechanisms induced by these risk factors, culminating in atrial fibrillation and endothelial dysfunction (**right**). Autonomic nervous system (ANS); renin–angiotensin–aldosterone system (RAAS). Parts of the figure were drawn using pictures from Servier Medical Art. Servier Medical Art by Servier is licensed under a Creative Commons Attribution 3.0 Unported License (https://creativecommons.org/licenses/by/3.0/, accessed on 8 February 2024).

**Figure 2 ijms-25-02317-f002:**
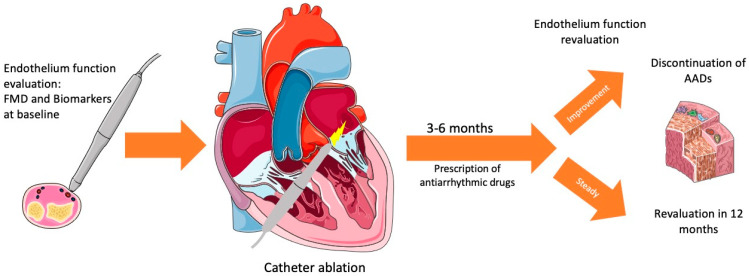
A possible approach for patients undergoing catheter ablation for atrial fibrillation. Parts of the figure were drawn using pictures from Servier Medical Art. Servier Medical Art by Servier is licensed under a Creative Commons Attribution 3.0 Unported License (https://creativecommons.org/licenses/by/3.0/, accessed on 2 January 2024).

## Data Availability

No new data were created or analyzed in this study. Data sharing is not applicable to this article.

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
