# Peer review of "Unveiling the Role of Endothelial Dysfunction: A Possible Key to Enhancing Catheter Ablation Success in Atrial Fibrillation"

_ijms, 2024, doi:10.3390/ijms25042317_

Round 1
Reviewer 1 Report
Comments and Suggestions for Authors
In the current review, Zakynthinos et al. focused on the complex interplay between endothelial dysfunction and the mechanism of atrial fibrillation (AF). In particular, the authors evaluated whether endothelial dysfunction, as assessed by biomarker levels or physiological measurements, might influence the success of AF catheter ablation. The paper provides a nice overview of the definition and assessment of endothelial dysfunction, examines the role of common cardiovascular risk factors in the pathogenesis of both AF and endothelial dysfunction, and sheds light on the reciprocal relationship between endothelial dysfunction and AF catheter ablation. The manuscript is well written, and the content is of clinical interest to the readership. I have only a few comments.
1) The authors emphasize the importance of a holistic and integrated approach to patients with AF to improve outcomes of catheter ablation and post-ablation management. Given the variety of endothelial function assessment methods described in the literature, which do the authors consider to be the most useful for use in routine clinical practice? In addition, although there is a paucity of data, newer energy sources such as pulsed field ablation may induce a less inflammatory response than conventional energy. Do the authors expect endothelial function to play a lesser role in this subgroup? Please comment.
2) In line 188, the authors state that no clinical trials have yet demonstrated the superiority of catheter ablation over AADs for the treatment of AF. However, although the two approaches have not yet reached the same level of recommendation in current guidelines, except for patients with HRrEF, some RCTs have demonstrated the superiority of the former in patients with paroxysmal AF (DOI: 10.1093/europace/euab029; 10.1056/NEJMoa2029980). Please comment.
3) Please consider adding a descriptive image that illustrates the complex interplay between common cardiovascular risk factors and the pathogenesis of both endothelial dysfunction and atrial fibrillation.
4) Please consider changing the word "evaluation" to "cardioversion" in line 89.
5) Please consider rephrasing the following sentence (line 128): “Increased sympathetic activity, resulting in sympathetic denervation and an imbalance in the autonomic nervous system, also plays a role in AF development in the heart”. In diabetic patients, increased sympathetic tone results from parasympathetic denervation. This is followed by sympathetic denervation.
6) Please consider changing the word “of” to “on” in line 197.
Comments on the Quality of English Language
None.
Author Response
Dear Reviewer,
We sincerely appreciate the time and effort you dedicated to reviewing our manuscript. Your expert feedback has been instrumental in guiding our revisions and enhancing the overall quality of our work. We are pleased to report that we have addressed each of your suggestions as follows:
-
Pulse Field Ablation: In response to your first suggestion, we revisited our discussion on pulse field ablation. Although we had initially mentioned this in our text, we have now expanded upon this information to provide a more comprehensive answer to your question. This revision enriches the manuscript's content and offers readers a clearer understanding of pulse field ablation's significance in the context of our study.
-
Randomized Controlled Trial (RCT) Evidence: Following your second suggestion, we have included a mention of an RCT that demonstrates the superiority of catheter ablation (CA) over alternative treatments. We have highlighted the importance of this finding and explicitly stated that it underscores the need for further improvements in catheter ablation techniques. The relevant reference has been added to support this addition, ensuring that readers have access to the foundational research behind our discussion.
-
Addition of a New Figure: In accordance with your third suggestion, we have added a figure to the manuscript. We believe this figure effectively illustrates the key points discussed and hope it aligns with your expectations. This visual addition aims to enhance readers' comprehension and engagement with the material presented.
We have also diligently addressed all other suggestions you provided, ensuring that each point was carefully considered and incorporated into our revisions. We believe these changes significantly improve our manuscript, making it a more valuable contribution to the field.
We are grateful for your constructive feedback and the opportunity to refine our work further. We eagerly anticipate your thoughts on the revisions and hope our manuscript now meets the high standards of your journal.
Thank you once again for your invaluable input and guidance.
Sincerely,
GE. Zakynthinos
Reviewer 2 Report
Comments and Suggestions for Authors
I read with great interest the paper “Unveiling the Role of Endothelial Dysfunction: A Key to En- 2 hancing Catheter Ablation Success in Atrial Fibrillation” by Zakynthinos et al.
The content is presented in a logical manner. English is fine.
Here are some suggestions for improvement:
1. line 30: please verify [1)]
2. Include transition sentences between paragraphs to smoothly guide the reader from one point to the next. It might also be useful to use subheadings.
3. Atrial fibrillation stands as the most prevalent arrhythmia, with its occurrence ranging from 0.1% in patients under 55 years to exceeding 9% in octogenarians (Medicina (Lithuania), Open Access, Volume 55, Issue 10, 2019, Article number 617). The elevated prevalence of atrial fibrillation in the elderly underscores the potential contribution of multiple comorbidities to endothelial dysfunction. Research indicates that this dysfunction can be linked to various conditions prevalent in the aging population, such as diabetes and heart failure (Medicina (Lithuania), Open Access, Volume 55, Issue 10, 2019, Article number 617, and Current Issues in Molecular Biology, Open Access, Volume 45, Issue 8, Pages 6651 - 6666, August 2023). Understanding the intricate association between atrial fibrillation, comorbidities, and endothelial dysfunction is crucial for devising comprehensive strategies in managing this complex clinical scenario. Please provide a paragraph explaining this possible association.
Author Response
Dear Reviewer,
We are writing to express our deepest gratitude for your thorough review and insightful suggestions regarding our manuscript. Your expertise and constructive feedback have been invaluable in enhancing the quality and readability of our work.
We are pleased to inform you that we have meticulously implemented all the recommended changes you suggested. Specifically, we have added subheadings throughout the manuscript to improve its organization and facilitate a smoother reading experience. Additionally, we have incorporated transition sentences between sections to ensure a coherent flow of ideas and arguments.
In response to your specific request, we have also added a new paragraph about AF, ED and comorbidities, such as ageing. We also added the references that you suggested.
We believe these revisions have significantly improved our manuscript, making it more structured, reader-friendly, and comprehensive. We are hopeful that these changes align with your expectations and enhance the manuscript to a level suitable for publication.
Thank you once again for your valuable contributions to the refinement of our manuscript. We eagerly await your feedback and hope our revisions meet your approval.
Sincerely,
GE. Zakynthinos
Reviewer 3 Report
Comments and Suggestions for Authors
The manuscript entitled Unveiling the Role of Endothelial Dysfunction: A Key to Enhancing Catheter Ablation Success in Atrial Fibrillation is a review that presents the connection between atrial fibrillation and endothelial dysfunction and how endothelial factors might influence the success rate of catheter ablation in this arrhythmia.
Minor revision
The manuscript is well and clear written. The authors made a synthesis about the effect of endothelial dysfunction on the recurrence of atrial fibrillation after catheter ablation (table 1) and the effect of catheter ablation on endothelial dysfunction (table 2). It is a good synthesis about the relationship between AF and endothelial dysfunction but I recommend underlying practical implications in key points. It is more useful to nominate how can we do preoperative evaluation of endothelial function, as well as the follow-up assessments for a period (how much) after the procedure. In addition, how should be monitored endothelial function in patients with atrial fibrillation in clinical practice? If these are not clearly established (which are the parameters and the timing) than the authors must specify this in conclusion. Also, the title could be changed by adding: A possible key to enhancing…. (not A key to enhancing…).
Please verify all acronyms. For example on page 3 line 99, AF is another one explained. Many acronyms are explain more than one.
Author Response
Dear Reviewer,
We would like to express our sincere gratitude for your insightful feedback and constructive suggestions regarding our manuscript. Your expertise has significantly contributed to the enhancement of our work.
We are pleased to inform you that we have carefully considered and implemented all the changes you suggested. These revisions have undoubtedly improved the clarity, accuracy, and overall quality of our manuscript.
Regarding the best methods for assessing endothelial function, we acknowledge the current uncertainty within the scientific community. As you rightly pointed out, there is no consensus on a gold standard method for evaluating endothelial function. In response to your valuable comment, we have included a discussion in the conclusions section of our manuscript. Here, we emphasize the lack of a universally accepted method and suggest that future research should aim to address this gap in the literature. This addition not only enriches our discussion but also highlights an important area for future investigation.
We hope that these revisions meet your approval and further strengthen our submission. We are committed to contributing valuable insights to the field and believe that our work, with your valuable suggestions incorporated, moves us closer to this goal.
Thank you once again for your thoughtful review and valuable contributions to the improvement of our manuscript. We look forward to your further comments and hope for a positive evaluation.
Sincerely,
GE Zakynthinos
Round 2
Reviewer 1 Report
Comments and Suggestions for Authors
I thank the authors for addressing all the issues raised. I have no further comments.